# Creation of an unexpected plane of enhanced covalency in cerium(III) and berkelium(III) terpyridyl complexes

Alyssa N. Gaiser [1], Cristian Celis-Barros [1], Frankie D. White[1], Maria J. Beltran-Leiva[1], Joseph M. Sperling[1], Sahan R. Salpage[1], Todd N. Poe [1], Daniela Gomez Martinez[1], Tian Jian[2], Nikki J. Wolford[3], Nathaniel J. Jones [1], Amanda J. Ritz[1], Robert A. Lazenby [1], John K. Gibson[2], Ryan E. Baumbach [4], Dayán Páez-Hernández[5], Michael L. Neidig [3] & Thomas E. Albrecht-Schönzart [1]✉

Controlling the properties of heavy element complexes, such as those containing berkelium, is challenging because relativistic effects, spin-orbit and ligand-field splitting, and complex metal-ligand bonding, all dictate the final electronic states of the molecules. While the first two of these are currently beyond experimental control, covalent M–L interactions could theoretically be boosted through the employment of chelators with large polarizabilities that substantially shift the electron density in the molecules. This theory is tested by ligating Bk[III] with 4′-(4-nitrophenyl)-2,2′:6′,2″-terpyridine (terpy*), a ligand with a large dipole. The resultant complex, Bk(terpy*)(NO$_3$)$_3$(H$_2$O)·THF, is benchmarked with its closest electrochemical analog, Ce(terpy*)(NO$_3$)$_3$(H$_2$O)·THF. Here, we show that enhanced Bk–N interactions with terpy* are observed as predicted. Unexpectedly, induced polarization by terpy* also creates a plane in the molecules wherein the M–L bonds trans to terpy* are shorter than anticipated. Moreover, these molecules are highly anisotropic and rhombic EPR spectra for the Ce[III] complex are reported.

[1] Department of Chemistry and Biochemistry, Florida State University, Tallahassee, FL 32306, USA. [2] Chemical Sciences Division, Lawrence Berkeley National Laboratory, Berkeley, CA 94720, USA. [3] Department of Chemistry, University of Rochester, Rochester, NY 14627, USA. [4] National High Magnetic Field Laboratory, Tallahassee, FL 32310, USA. [5] Center for Applied Nanosciences, Universidad Andres Bello, República 275, Santiago, Chile. ✉email: talbrechtschoenzart@gmail.com

Unexpected properties, structures, and reactivities emerge in heavy elements because their high nuclear charge accelerates surrounding electrons to relativistic speeds, altering orbital shapes and energies and the nature of chemical bonds[1]. This in turn, leads to abrupt changes in behavior between neighboring elements[2–8] and a breakdown of simple descriptions of electronic structure that can be used to explain emerging properties[1,9–17]. Examples of these discontinuities include the large volume expansion between α-Pu and α-Am, and the corresponding localization of $5f$ electrons that leads to superconductivity in α-Am at low temperatures[18], as well as the diminishment of redox activity that occurs at this same juncture in the actinide series[19]. Moreover, between berkelium and californium a second transition occurs whereby the divalent state becomes metastable in both the pure elements and in compounds[2,15]. Understanding the origin of these step functions between neighboring actinides has been at the forefront of research since the dawn of the Atomic Age.

In a more general sense, many electronic factors arise in magnitude in a nonlinear manner in heavy elements. For example, between hydrogen ($Z = 1$) and bismuth ($Z = 83$) there is only a 25% increase in the relative mass of the $1s$ electrons induced by acceleration afforded by nuclear charge. In contrast, between bismuth and uranium ($Z = 92$), the perceived mass increases by an additional 25% even though $Z$ has only increased by 9[20]. Spin–orbit coupling, a consequence of relativistic effects, scales as $Z^4$[1], and is large enough in magnitude to mix $L$ and $S$ states together in the traditional Russell–Saunders coupling scheme[21]. Moreover, the spin–orbit splitting not only affects the ground state but also the excited states. In the actinide series, the splitting is large enough to mix ground and excited configurations giving rise to multi-reference states[19,22,23]. In $Bk(IO_3)_3$, for example, the ground state consists of ~70% the $LS$ term ($^7F_6$) and ~30% the first excited state ($^5G_6$). Thus, the magnetic properties of $Bk(IO_3)_3$ would be expected on this basis alone to differ from the ostensibly isoelectronic $Tb^{III}$ analog, and this is observed[6]. Similar differences are found between $Bk(Hdpa)_3$ and $Tb(Hdpa)_3$ (dpa = dipicolinate; 2,6-pyridinedicarboxylate)[8]. In addition to magnetic susceptibility, optical properties and even bond lengths not only differ between formally isoelectronic ions (e.g., $Dy^{III}$ and $Cf^{III}$) but also between neighboring actinides in a non-systematic way as observed in the aforementioned breaks between plutonium and americium and again between berkelium and californium[2–8].

In actinide compounds, the frontier orbitals ($5f$, $6p$, $6d$, $7s$, $7p$) can contribute to bonding to a greater extent than occurs in corresponding lanthanide systems despite $f$-element–ligand bonds being dominated by electrostatic interactions[24–26]. This can also lead to deviations in chemical and physical properties between the $4f$ and $5f$ series that manifests in the adoption of different structures with distinct physical properties emerging[3]. It is also now established that ligand-field splitting is larger than expected beyond curium, and examples in both berkelium and californium systems exist where this splitting is ca. 2000 cm$^{-1}$[5,6]. Coupling these features together with the decreased e$^-$···e$^-$ repulsion between $5f$ electrons vs. those in $4f$ orbitals[27,28] leads to the so-called intermediate coupling regime where no single electronic effect (interelectronic repulsion, ligand field, spin–orbit coupling) dominates, and predicting the physico-chemical properties of actinide molecules becomes quite challenging[2,6,19].

Thus, the question arises as to whether the electronic structures of actinide complexes can be substantially altered through the design of specific electronic attributes of the ligands surrounding it given the complexities of the metal ions in these systems. While there are certainly numerous examples of the use of ligands to create specific symmetries[29,30], large binding constants[31,32], and open-coordination sites around actinides that lead to unique

reactivities[12,33–39], substantial changes in bonding might also be achievable in actinide complexes by using ligands that create large dipole moments. Guidance on how to achieve this effect exists from the large body of work for designing organic nonlinear optical materials[40]. Herein, we show that a terpyridyl ligand with a large polarizability, 4'-(4-nitrophenyl)-2,2':6',2''-terpyridine (terpy*), can be used to create unusual bonding and rare spectroscopic features in a berkelium(III) complex. To the best of our knowledge, this is only the sixth berkelium compound for which a single crystal structure has been solved, thus the opportunity to compare it to its closest electrochemical analog, $Ce^{III}$, was also undertaken in this work.

## Results and discussion

**Synthesis.** $^{249}Bk$ has a half-life of 330 days and therefore has an unusually high specific activity. This is especially apparent when compared to earlier actinide isotopes such as $^{238}U$ that possesses $t_{1/2} = 4.5 \times 10^9$ years. Even a few milligrams of $^{249}Bk$ creates Ci levels of radiation. Recoil from the β decay of $^{249}Bk$ is in the keV range and creates local disruption of chemical bonds. Moreover, its rapid decay to $^{249}Cf$ ($t_{1/2} = 351$ years) creates an α emitter with energies above 5 MeV that again leads to further sample destruction. Substantial degradation of solvents, ligands, and compounds occurs within a few days because, in addition to the damage from nuclear recoil, and the damage paths from the trajectories of the α and β particles, reactive radiolytic products, such as hydroxyl radical, create undesirable side reactions that yield intractable mixtures of products. Crystals of targeted compounds must therefore be grown, isolated, and fully characterized within hours of preparation or Coulombic explosions occur that render them into nanocrystalline or amorphous solids that are difficult to characterize further.

$^{249}Bk$ decays to $^{249}Cf$ at a rate of ~1.5% week$^{-1}$. This necessitates the separation of $^{249}Cf$ from $^{249}Bk$ immediately prior to synthesis. $^{249}Bk$ was isolated from an aged mixture of $^{249}Bk/^{249}Cf$ that had a ratio of ca. 1:5 via the oxidation of $Bk^{III}$ to $Bk^{IV}$ under slightly basic conditions using 30% $H_2O_2$. This vigorous reaction results in the precipitation of $Bk(OH)_4$ as a deep red solid and leaves $Cf^{III}$ behind as an emerald green solution. This product was subsequently converted to $Bk(NO_3)_3 \cdot nH_2O$ by gentle fuming in 8 M $HNO_3$.

The reaction of freshly-prepared $Bk(NO_3)_3 \cdot nH_2O$ or $Ce(NO_3)_3 \cdot nH_2O$ with 4'-(4-nitrophenyl)-2,2':6',2''-terpyridine (terpy*) in tetrahydrofuran (THF) yields golden columns of $Bk(terpy*)(NO_3)_3(H_2O) \cdot THF$ and $Ce(terpy*)(NO_3)_3(H_2O) \cdot THF$ (**Bk1** and **Ce1**, respectively) within a few hours. The corresponding $Ce^{III}$ complex lacking the appended 4-nitrophenyl group, $Ce(terpy)(NO_3)_3(H_2O) \cdot THF$ (**Ce2**), was also synthesized for comparison by similar methods. Further synthetic details for **Bk1**, **Ce1**, and **Ce2** can be found in the Supplementary Information in the "Supplementary Methods" section.

**Structural characterization.** Single crystal X-ray diffraction data from crystals of **Bk1**, **Ce1**, and **Ce2** were measured from samples cooled to 28 K using a helium cryostat. While such data collections are fraught with technological woes, such as rapid and severe icing, they potentially allow for significant improvements in the precision of bond distances (by an order of magnitude), increased diffraction intensities, and reduced thermal motion of atoms[41]. The latter reduction means that the measured bond distances have substantially less libration[42,43] and are much closer to libration-free interatomic distances.

**Bk1** and **Ce1** are isomorphous and adopt the same structure as found with other trivalent lanthanides and actinides as we recently reported for $Am^{III}$[44]. The structure of Bk(terpy*)

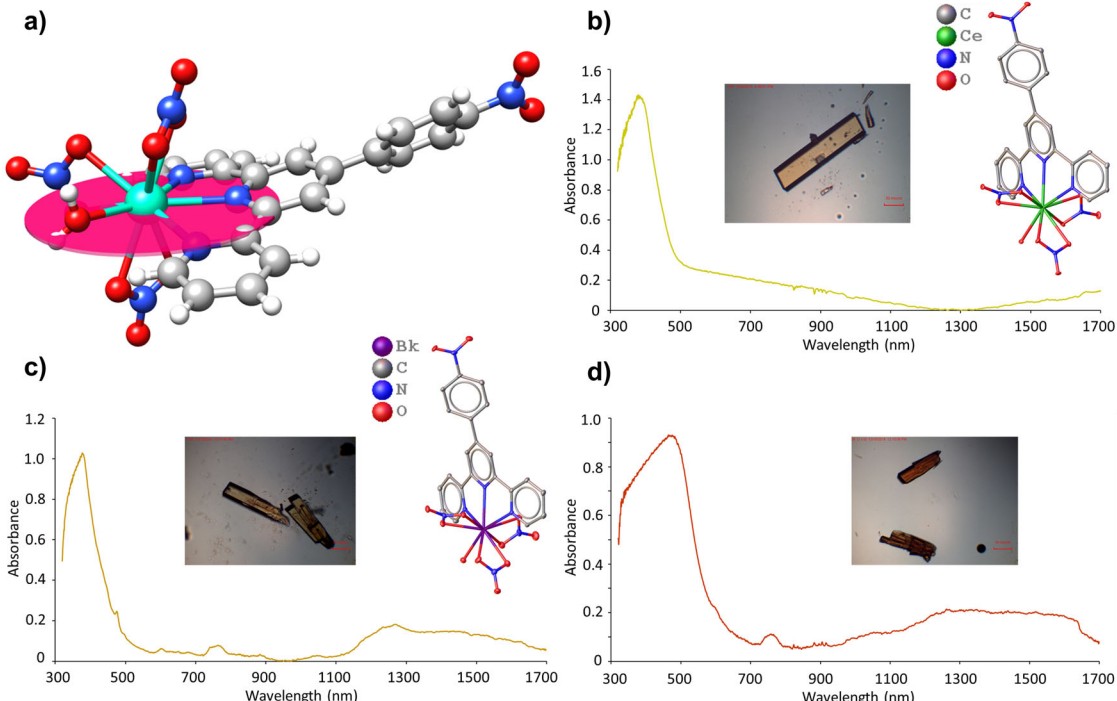

**Fig. 1 Characterization of Ce$^{III}$ and Bk$^{III}$ terpyridine complexes. a** Depiction of the plane defined by the terpyridine derivative, water molecule, and equatorial nitrate ligand. The optimization of the plane was performed using the coordinates of the terpyridine nitrogen atoms, metal center, and water oxygen atom. **b** Absorption spectrum, crystal structure, and single crystal picture of **Ce1**. **c** Absorption spectrum, crystal structure, and single crystal pictures of **Bk1** the day they were collected and **d** 3 days later.

$(NO_3)_3(H_2O)$ with the co-crystallized THF molecule omitted is shown in Fig. 1a and contains a Bk$^{III}$ cation bound by three bidentate nitrate anions, one tridentate terpy* ligand, and one water molecule yielding a ten-coordinate environment. These bond distances are tabulated in the Supplementary Materials, but some critical features are noted below. They can also be used to calculate the ionic radius of ten-coordinate Bk$^{III}$, of which this is the first example, and yield a value of 1.19 Å that parallels that of Sm$^{III}$[44].

The key feature of the M(terpy*)$(NO_3)_3(H_2O)$ (M = Bk, Ce) molecules is that they contain a nearly planar moiety composed of terpy*, the bound water molecule, and one of the nitrate molecules (Fig. 1a). This nitrate anion is bisected by the polarization plane. Two additional nitrate anions that bind the metal centers are also present above and below this plane. The simplest way to illuminate the differences between ligands in this plane vs. those out of the plane is achieved by comparing the asymmetry of Bk-O bond lengths with the nitrate anions. In **Bk1**, the deviation between the two Bk-O bond lengths of the nitrate anions are 0.088(3) and 0.062(3) Å above and below the plane, respectively; whereas the Bk-O bond lengths to the nitrate anion trans to the terpy* are more similar and differ by 0.023(3) Å. Similarly, in **Ce1** the differences in the Ce-O bond lengths of the nitrate anions above and below the plane of the terpy* are 0.031(2) and 0.034(2) Å; while the difference between the Ce-O bond lengths of the nitrate molecule trans to the terpy* is 0.013(2) Å. For **Bk1**, this gives an average difference of 0.075(3) Å axially and 0.023(3) Å in the plane. Likewise, **Ce1** gives an average difference on 0.033(2) Å axially and 0.013(2) Å in the plane.

A similar observation is made when examining the M-OH$_2$ bond distances in these molecules. Here a comparison is made between the structure of **Ce1** and **Ce2** (Supplementary Fig. 17) where the latter lacks the 4-nitrophenyl moiety. In **Ce2**, the water molecule is not co-planar with the terpy ligand and the Ce-OH$_2$

bond distance is 2.5267(6) Å. In contrast, in both **Bk1** and **Ce1** the bound water molecule is co-planar with the terpy* ligand, and the Ce-OH$_2$ bond is statistically shorter (3σ) than found in **Ce2** with a distance of 2.491(2) Å. The difference in conformations between **Bk1/Ce1** and **Ce2** is likely a consequence of the polarization by terpy* (vide infra). **Bk1**, **Ce1**, and **Ce2** all contain an outer-sphere THF molecule that interacts with the bound water molecule through hydrogen bonding. The disparate placement of the water molecule in **Ce2** could alternatively be attributed to crystal packing, as it lacks the nitrophenyl group present in **Ce1** and **Bk1**.

**Gas-phase studies**. To further understand the strength of the interaction of Ce$^{III}$ with terpy and terpy*, collision-induced dissociation of gas-phase coordination complexes was carried out ("Additional Discussion" section in Supplementary Information). These studies were compared to the previously reported **Eu1** structure[44]. Our results reveal that in gas-phase complexes both terpy and terpy* bind more strongly to Ce$^{III}$ than Eu$^{III}$ and that both Ce$^{III}$ and Eu$^{III}$ bind more strongly to terpy than terpy*. This measurement is consistent with the terpy* being a weaker σ-donor than terpy as would be anticipated from the electron-withdrawing nature of the 4-nitrophenyl group. It is noteworthy that these results cannot be correlated to the formation of a plane of interaction due to the crystal packing effects vs. gas-phase molecular geometries. Distinctive processes corresponding to oxidation to Ce$^{IV}$ and reduction to Eu$^{II}$, which directly reflect condensed-phase redox properties, are revealed upon dissociation of gas-phase complexes.

**Optical and magnetic circular dichroism (MCD) spectroscopy**. Ultraviolet–visible (UV-vis)–near-infrared spectra were collected from single crystals of **Ce1** and **Bk1**, and again for **Bk1** 3 days

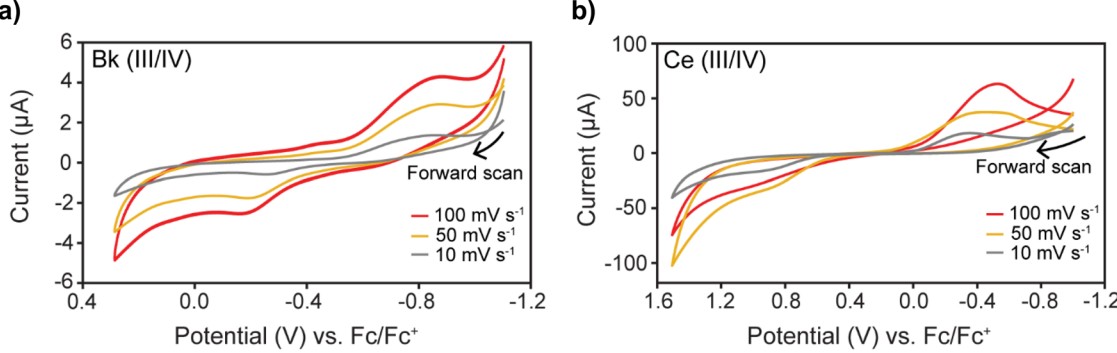

**Fig. 2 Cyclic voltammograms of Ce$^{III}$/Ce$^{IV}$ and Bk$^{III}$/Bk$^{IV}$ terpyridine derivatives in THF and electrochemical data.** Potential scan rates of 10 (gray), 50 (yellow), and 100 (red) mV s$^{-1}$ were used. **a** Voltammograms showing the oxidation (III/IV) and reduction (IV/III) of the Bk/Terpy* complex, in 0.1 M BArF (supporting electrolyte). **b** Voltammograms showing the oxidation (III/IV) and reduction (IV/III) of the Ce/Terpy complex, in 0.1 M TBA PF$_6$ (supporting electrolyte). All data have been corrected to zero volts vs. an internal Fc/Fc$^+$ reference.

**Table 1 Electrochemical data.**

| Complex | $E_{1/2}$ (vs. Fc/Fc$^+$) (V) | $E_{p,a}$ (vs. Fc/Fc$^+$) (V) | $E_{p,c}$ (vs. Fc/Fc$^+$) (V) | $E_{p,a} - E_{p,c}$ (V) | $\Delta E_{p,c}$ for complex vs. non-complexed (V) |
|---|---|---|---|---|---|
| Ce1 | 0.249 | 1.020 | −0.522 | 1.542 | −0.277 |
| Bk1 | 0.207 | −0.158 | −0.887 | 0.729 | N/A |

All data have been corrected to zero volts vs. an internal Fc/Fc$^+$ reference and was recorded at 100 mV s$^{-1}$.

after the crystals were formed (Fig. 1b–d). All three spectra show a broad band centered near 400 nm that is assigned to intra-ligand transitions for the terpy* complexes. For **Bk1**, the characteristic *f–f* transitions for Bk$^{III}$ are observed[45], confirming that Bk$^{III}$ has not been oxidized to Bk$^{IV}$ (Fig. 1c, d).

To aid in the assignment of the absorption spectrum of **Ce1**, the 5K MCD spectrum was obtained in the UV-vis region (Supplementary Fig. 16). *C*-term MCD spectroscopy provides higher resolution and the benefit of both positive and negative sign transitions that aids in separating and assigning overlapping transitions like those observed in **Ce1**. The spectrum could be fit to multiple transitions, as predicted by computational analysis that were subsequently assigned to a series of $4f \longrightarrow$ ligand and ligand $\longrightarrow 4f$ charge–transfer transitions (Supplementary Table 5).

**Cyclic voltammetry of Ce1 and Bk1.** The **Ce1** and **Bk1** complexes exhibit similarly quasi-reversible electrochemical behavior, with very wide peak separations evident in the cyclic voltammograms (CVs) in Fig. 2. The **Bk1** complex was more reversible than the **Ce1** complex, as demonstrated by the smaller peak-to-peak separation (0.729 vs. 1.542 V, Table 1). A variety of cerium complexes have previously been shown to exhibit low reversibility[46], and varying degrees of quasi-reversibility have been exhibited in selected lanthanide cryptates in THF in a previous study[47]. Cerium undergoes a potential shift of ~300 mV upon complexation with terpy* (Table 1) when compared to cerium nitrate (i.e., prior to **Ce1** complex formation). The cerium and berkelium complexes had (IV/III) reduction peak potentials, $E_{p,c}$, of −0.522 and −0.887 V, respectively, differing by about 0.350 V, and (III/IV) oxidation peak potentials, $E_{p,a}$, of 1.020 and −0.158 V, respectively, differing by 1.180 V. The very wide peak-to-peak separations, an indication of poor reversibility, are much greater than those found in previous work[5,46].

All voltammograms showed the presence of water, for which no effort was made to remove, since water is coordinated to the metal center of the complex. The sharp rise in anodic current at the end of the forward sweep is indicative of water oxidation, and the initially large cathodic current can also be explained by the reduction of water. Since the amount of water in the sample is likely to have varied between experiments, these currents were also variable. The anodic current was close to the oxidation of Ce$^{III}$ to Ce$^{IV}$, making the peak less prominent. The solvent window of water is much smaller than that of THF, so the presence of water posed a challenge for the observation of the anodic peak[48]. Peak identity was confirmed using different concentrations of the complex, with all other variables kept the same (Supplementary Information Fig. 19). This result also confirmed that, as well as increased complex concentration, the addition of more complex also resulted in more water present in the solution.

A potential scan rate-dependent current response was observed for the voltammetry, as expected when using a macroelectrode. CVs were recorded at 10, 50, and 100 mV s$^{-1}$, which resulted in increasing current magnitudes for both the reduction and oxidation peaks of **Ce1** and **Bk1** (Fig. 2 and Table 1). Lower scan rates resulted in more clearly defined oxidation peaks, with lower current. These data were collected for freshly prepared complexes in THF solution. Data were collected at similar times after complex formation, to minimize the effects of solvent loss due to evaporation, which would increase peak current magnitude.

**Electron paramagnetic resonance (EPR) and magnetism.** Ce$^{III}$ has a $4f^1$ configuration that has been shown to exhibit aniso-tropic EPR spectra in multiple molecular systems[49,50]. According to the 5K EPR spectrum of a powder sample of **Ce1** (Fig. 3a), Ce$^{III}$ displays an anisotropic signal with three distinct *g* values, 2.7, 1.1, and 0.6. This anisotropy reflects the different binding of the terpy*, aquo, and nitrate ligands to Ce$^{III}$ resulting in three distinct molecular axes and a rhombic system.

The *f*-block complexes commonly exhibit large deviations of the *g*-factors from the spin only value ($g_e$ ~ 2) and pronounced magnetic anisotropy. These effects are produced by an orbital contribution to the magnetic moment that result from the spatial degeneracy of an open shell in combination with spin–orbit

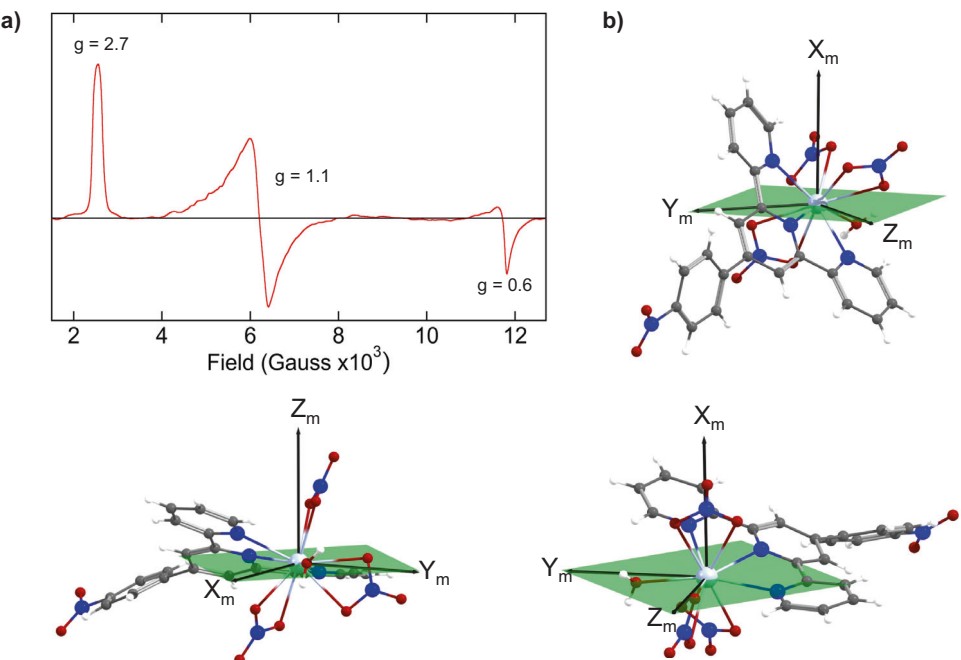

**Fig. 3 Electron paramagnetic resonance and *g*-factors for Ce1. a** Experimental EPR spectrum and the corresponding *g*-factors. **b** Quantization axes and magnetization planes for the three KDs ordered by increasing energy from left to right.

**Table 2 *g*-factors for Ce1.**

| $E$ (cm$^{-1}$) | $g_x$ | $g_y$ | $g_z$ | $\langle L_z \rangle$ | $\langle S_z \rangle$ | $\langle L_x \rangle$ | $\langle S_x \rangle$ | $\langle L_y \rangle$ | $\langle S_y \rangle$ |
|---|---|---|---|---|---|---|---|---|---|
| 0.0 | 0.789 | 1.372 | 2.671 | −1.887 | 0.275 | −0.409 | 0.007 | −0.990 | 0.152 |
| 131.0 | 0.378 | 0.969 | 2.535 | −1.708 | 0.219 | −0.254 | 0.032 | −0.601 | 0.058 |
| 299.7 | 2.640 | 1.787 | 0.859 | −0.628 | 0.099 | −1.817 | 0.248 | −1.139 | 0.123 |

Theoretical energies, *g*-factors, and the orbital and spin angular momentum expectation values obtained from spin–orbit CAS(1,7) wave functions for the three Kramer's doublets (KDs) derived from the $^2F_{5/2}$ ground multiplet.

interaction and covalent interactions with the ligands. Owing to the effect of the ligand field, the $^2F_{5/2}$ ground multiplet of Ce$^{III}$ splits into three Kramers doublets (KDs) characterized by a pseudospin $\mathbf{S} = 1/2$. This approximation refers to a spin acting in a model space of eigenfunctions $| M_S \rangle$ for the pseudospin projection onto a quantization axis that is useful to interpret our results. The calculated energies and *g*-factor components of these three KD states are presented in Table 2.

The calculated *g*-factors for the ground state agree with the experimental values (Fig. 3a and Table 2), given the quantization axes shown in Fig. 3b. An important contribution of angular momentum was observed for the three KDs that also have an opposite sign to the spin contribution as expected for an $4f^1$ configuration (less than half-filled shell) (Table 2). It is interesting to note that the *g*-factors describe a magnetization plane for KD1 (*yz*), KD2 (*yz*), and KD3 (*xy*), where *x*, *y*, and *z* represent the quantization axes for each KD. Furthermore, this is accompanied by significant contributions from the components of the orbital angular momentum defined on these planes of magnetization. A more detailed analysis (Table 2) shows that these planes are formed by the water molecule, the terpy* nitrogen atoms, and the equatorial nitrate group. The observation of this magnetic plane along with the significant angular momentum contribution to the *g*-factor may be related to the presence of a plane of covalency between Ce$^{III}$ and the ligands sitting on this plane[51], though further studies would be required to confirm this. On the other hand, the shape of the 4*f* electron density that is directly related to the occupation of the 4*f* natural orbitals shows an oblate nature, where the electron density is distributed preferentially in the plane (KD1 in Supplementary Fig. 3). This can be correlated with the spin magnetization that is distributed equally in the magnetization plane with an oblate shape. Since the total splitting of the $^2F_{5/2}$ ground term into $M_J$ substates (~300 cm$^{-1}$) matches the same magnitude as kT at room temperature (~210 cm$^{-1}$), the three KDs are populated (according to the Boltzmann distribution), and therefore all have a direct influence in the observed magnetic properties at room temperature.

For **Bk1**, the ground state corresponds to a non-KD derived from the ground multiplet $^7F_6$ according to the Bk$^{III}$ free ion with contributions from the excited multiplet $^5G_6$ (9%) due to spin–orbit coupling. This enables us to analyze this state using a pseudo-spin ½ Hamiltonian. Unlike **Ce1**, this ground state exhibits a large magnetic anisotropy with $g_z = 17.7$ and $g_x = g_y = 0.0$, owing to an important contribution of angular momentum ($L_z = 2.858$) and spin ($S_z = 2.993$) that is expected for a $5f^8$ configuration (more than half-filled shell). At room temperature, the calculated magnetic moment is 9.677 $\mu_B$, close to the expected value for a pure $^7F_6$ multiplet 9.72 $\mu_B$. The predicted magnetic susceptibility ($\chi T$) that reaches a value of 11.39 cm$^3$ K mol$^{-1}$ at room temperature decreases slowly even at low temperatures because of contributions of low-lying states. This behavior does not differ significantly from that previously observed in other Bk$^{III}$ compounds[6].

**Examination of chemical bonding.** From the molecular orbital perspective, two main aspects are to be emphasized to elucidate

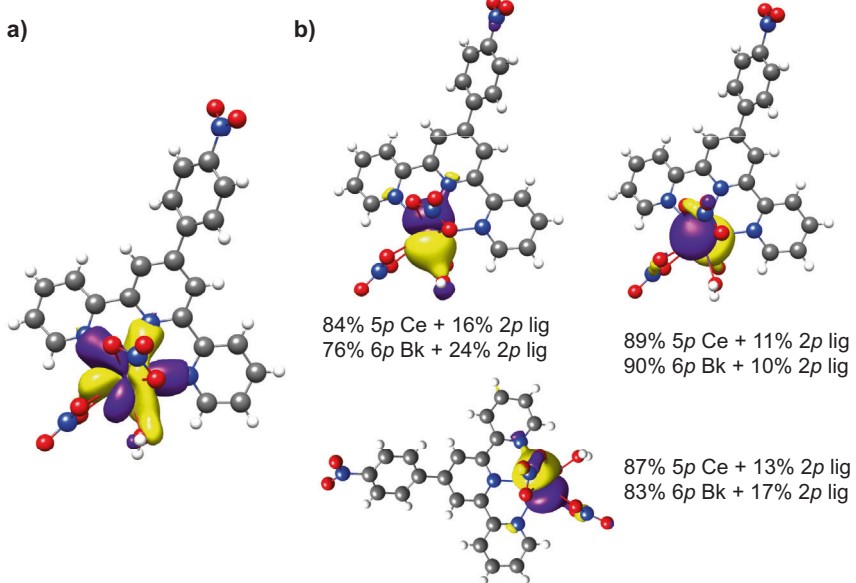

**Fig. 4 CASSCF natural orbitals and ligand-field DFT parameters.** Natural orbitals derived from state-specific scalar-relativistic CAS(1,7) and CAS(8,7) for **Ce1** and **Bk1**. **a** For both complexes, there is an f-orbital similar to the $f_{\pm3}$ orbital that shows a preferential orbital interaction between the metal center, terpy*, and the water molecule. This orbital is crucial to define the plane of covalency. **b** Orbital mixing between the 5p/6p orbitals with 2p-ligand orbitals showing the participation of the core in chemical bonding.

**Table 3 Ligand-field DFT parameters.**

| 5p parameters | $Ce^{3+}$ | Ce1 | Reduction | 6p parameters | $Bk^{3+}$ | Bk1 | Reduction |
|---|---|---|---|---|---|---|---|
| $F^0$ (5p,5p) | 14.23 | 8.23 | 42% | $F^0$ (6p,6p) | 14.51 | 7.24 | 50% |
| $F^2$ (5p,5p) | 7.74 | 4.45 | 42% | $F^2$ (6p,6p) | 8.03 | 3.99 | 50% |
| $\zeta_{SO}$ (5p) | 1.75 | 1.31 | 25% | $\zeta_{SO}$ (6p) | 6.62 | 4.60 | 31% |

Slater–Condon inter-electron repulsion and effective spin–orbit coupling parameters of the 5p and 6p shells obtained for the $Ce^{III}$ and $Bk^{III}$ free ions and **Ce1** and **Bk1** along with their corresponding reductions due to covalent interactions.

the nature of this plane of enhanced interactions: (i) the observation of one f orbital featuring the interaction with terpy* and aquo ligands (Fig. 4a), and (ii) the role of the 5p/6p semi-core orbitals in bonding to expose the 4f/5f shells. The latter is exemplified in the mixing of these metal orbitals with 2p-ligand orbitals (Fig. 4b). It is important for the reader to note that these orbital interactions represent subtle effects compared to the dominant force in the bond formation, i.e., electrostatic interactions. Further discussion on the electronic structure is found in the Supplementary Information (see "Theory" in the "Additional Discussion" section).

Ligand-field density functional theory[52] was used to evaluate the expansion of the 5p/6p radial functions through the reduction of the inter-electron repulsion for the **Ce1** and **Bk1** complexes with respect to their corresponding free ions[53]. Our results show that semi-core 5p/6p electrons are involved in covalent interactions due to the observed reduction in the inter-electron repulsion as well as the effective spin–orbit coupling parameter. The more polarizable character of the 6p shell in $Bk^{III}$, than the 5p in $Ce^{III}$, is evidenced in the increased reduction observed in **Bk1** (50% in $F^k$ and 31% in $\zeta_{SO}$) compared to **Ce1** (42% in $F^k$ and 25% in $\zeta_{SO}$) (Table 3). These results show that electron repulsion between semi-core electrons is overcome by the covalent interactions with the coordinating ligands. Furthermore, the involvement of the semi-core orbitals have been associated previously with the inverse trans influence (ITI)[54–56] that is now offered in a broader sense to understand covalency. The difference resides in that for

ITI the semi-core 6p orbitals "push from below" by hybridizing with the 5f-orbitals, whereas in **Ce1** and **Bk1** this occurs by direct mixing (i.e., hybridization) with the ligand 2p orbitals.

To shed light on this plane of covalency, the quantum theory of atoms in molecules (QTAIM)[57,58] was used to map the electron density at the interatomic region and derive useful metrics, such as delocalization indices (pairs of shared electrons), energy densities, and ellipticities. These metrics are helpful to describe the nature of the bond in terms of concentration of electron density $\rho(r)$ at the so-called bond critical point (BCP) (see "Additional Discussion" section in the Supplementary Information).

It is well known that trivalent lanthanides are considered to be hard Lewis acids or at least harder than actinides. Therefore, it should be expected that terpy* would bind significantly more strongly to $Bk^{III}$ than to $Ce^{III}$, which is not the case. Although it is possible to see a difference in M–N bond metrics, the most striking difference is observed for both M–$OH_2$ bonds. From Carnall's work on the spectroscopy of lanthanides and actinides, the aquo complexes are considered a "diluted ion," thus implying that their properties should resemble those of the free ion[59,60]. This approximation does not hold for **Ce1** and **Bk1**, where their M–aquo bonds are shown to display more significant covalent interactions and their $\rho_{BCP}(r)$ values are approximately of the same order as that of the metal–terpy* bonds (Fig. 5b and Supplementary Tables 6 and 7). It is important to note that differences in the accumulation of $\rho_{BCP}(r)$ are a direct measure of

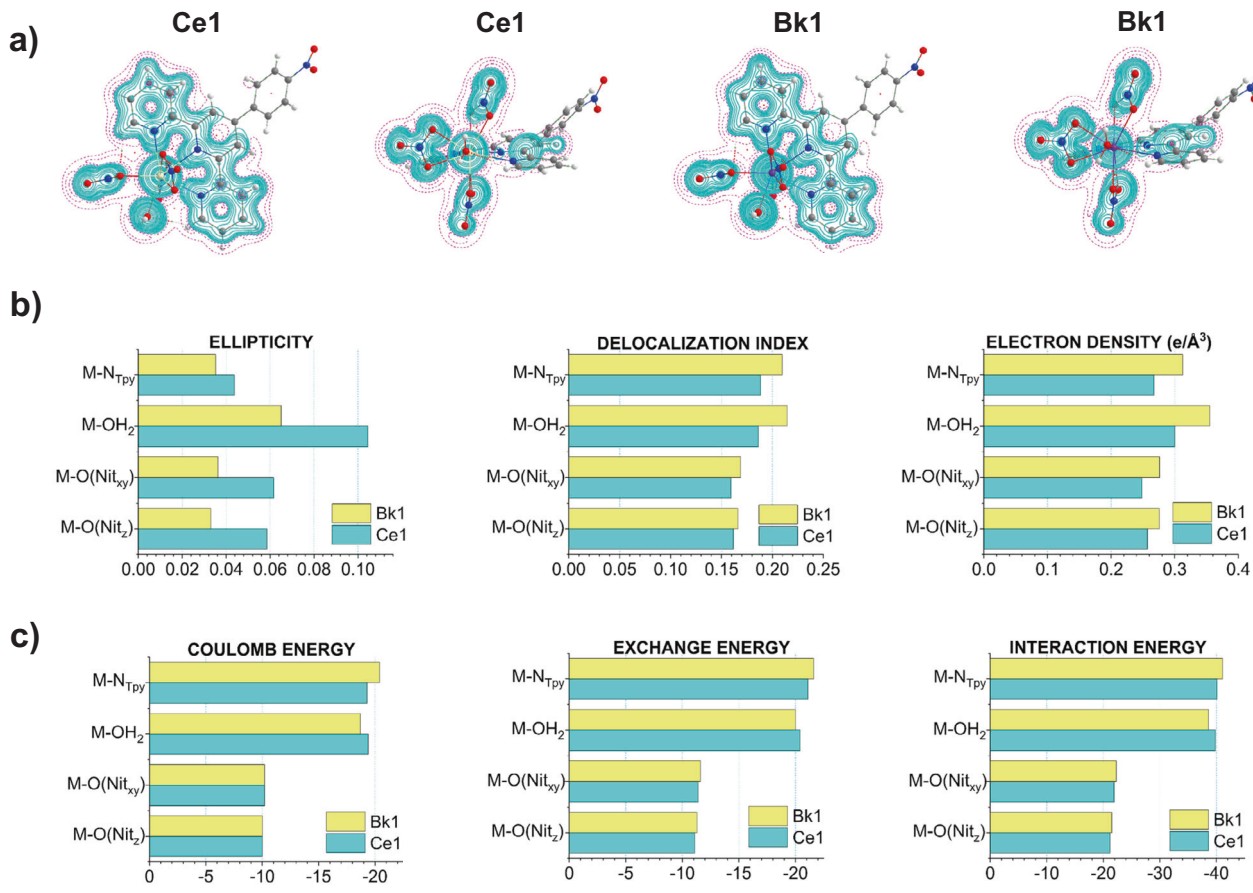

**Fig. 5 Bonding features of the plane of covalency based on the CASSCF electron densities. a** Plots of the total energy density, $H(r)$, in the preferential plane of interaction for **Ce1** (Ce = yellow sphere) and **Bk1** (Bk = purple sphere) and perpendicular to this plane. The solid cyan lines denote the regions where $H(r)$ is negative (covalent character), while pink dashed lines represent areas where $H(r)$ is positive (purely ionic). The water molecule as well as the terpy* N atoms display covalent interactions with the metal centers in both **Ce1** (cyan bars) and **Bk1** (yellow bars) compared to the nitrate ligands. **b** QTAIM metrics for **Ce1** and **Bk1**; the ellipticity describes the deviations from a cylindrical single bond as values differ from zero, while delocalization indices and electron densities describe the shared pairs of electrons and accumulation of electrons in the bond critical points, respectively. **c** Interacting quantum atom (IQA) energy decomposition analysis in kJ mol$^{-1}$. The total energy of interaction is decomposed into Coulomb or electrostatic and exchange or covalent energy components. M–N$_{terpy}$ correspond to average metrics of the three metal–terpy* bonds; M–O(Nit$_{xy}$) and M–O(Nit$_z$) refer to metal–nitrate bonds in the plane (equatorial) and out of the plane (axial), respectively. Tables with detailed information of QTAIM and IQA can be found in the Supplementary Information.

the strength of the bond, and therefore the orbital overlap between the two atoms involved. Despite this, only for **Bk1** this interaction is shown to be significantly covalent based on the $H(r)$ values (Supplementary Tables 6 and 7), while **Ce1** displays a negative, but close to zero, value. The formation of the plane of covalency is qualitatively shown in Fig. 5a, where the solid cyan lines represent regions where the total energy density is negative. To highlight this increased covalent character on **Ce1** and **Bk1**, we have previously reported the **Eu1** and **Am1** structures where the Eu–OH$_2$ bond displays a positive $H(r)$ value and the Am–OH$_2$ ca. $-5$ kJ mol$^{-1}$ Å$^{-3}$[44]. In contrast, **Ce1** and **Bk1** predict more excess of potential energy density, and therefore covalent character, with values $-1.8$ and $-22.8$ kJ mol$^{-1}$ Å$^{-3}$ (Supplementary Tables 6 and 7).

An alternative to the QTAIM approach is the interacting quantum atom (IQA) method[61] that provides an estimation of the interaction without employing the concept of BCP by integrating the electron density and providing a scheme of energy decomposition based on QTAIM. In principle, this natural partition of the molecule provides a more reliable estimation of the interacting densities between the metal and the ligands. However, we cannot fully rely on these numbers due to the

approximation introduced to calculate the two-electron interactions (Supplementary Information). Figure 5c shows the decomposition of the M–L interaction into Coulombic (electrostatic) and exchange (covalent) components (Supplementary Table 8). The latter directly relates to the strength of the interaction and therefore is the parameter to consider. The results overall agree with the QTAIM metrics except for the description of the M–OH$_2$ interaction that suggest the Ce–OH$_2$ interaction to be stronger than the Bk–OH$_2$ bond (Fig. 5c). This difference in strength is rather unexpected from the increased covalency one would expect for actinides over lanthanides. The origin could reside in the fact that lanthanides are more oxophilic than actinides, and therefore Bk$^{III}$ generally displays a preference toward N-donor ligands, whereas Ce$^{III}$ toward O-donor molecules. Regardless, all of our theoretical results support that the unusual behavior of the water molecule is attributed to the effect of the terpy* ligand on the metal center that causes the formation of a preferential plane of covalency.

In summary, the combination of structural, spectroscopic, electrochemical, and theoretical analysis of M(terpy*) (NO$_3$)$_3$(H$_2$O) (M = Bk, Ce) all support that the large polarizability of the terpy* ligand creates a plane where M–L

interactions are enhanced and a highly anisotropic electronic environment around the metal centers exists. More generally, **Bk1** shows greater involvement of the frontier orbitals in forming chemical bonds than occurs in **Ce1**, and this in turn is reflected in improved quasi-reversibility of electrochemical processes. These compounds represent proof of concept that the principles used to guide the synthesis of organic nonlinear optical materials, i.e., donor–acceptor molecules, can also be used to create ligands that enhance the involvement of frontier orbitals in forming chemical bonds in the 5*f* series. A large and diverse family of compounds should be achievable.

## Data availability

The data that support the findings of this study have been deposited in the CCDC database with the accession codes 1857536, 2050447, and 2050448 that contain the supplementary crystallographic data for this paper. These data can be obtained free of charge via www.ccdc.cam.ac.uk/data_request/cif, or by emailing data_request@ccdc.cam.ac.uk, or by contacting The Cambridge Crystallographic Data Centre, 12 Union Road, Cambridge CB2 1EZ, UK; fax: +44 1223 336033.

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

## Acknowledgements

We thank the Radiation Safety and Control personnel, Jason Johnson and Ashley Gray at Florida State University for their oversight during these challenging experiments. We also thank Xingsong Lin for assistance in powder X-ray diffraction experiments. This work has been supported by the U.S. Department of Energy, Office of Basic Energy Sciences, Heavy Elements Chemistry Program under Award Numbers DE-FG02-13ER16414 (to T.E.A.-S. for synthetic, crystallographic, spectroscopic, and electrochemical studies) and DE-AC02-05CH11231 (to J.K.G. for gas phase experiments), and U.S. Department of Energy, Office of Science, Early Career Research Program under Awards DE-SC0016002 and DE-SC0021917 (to M.L.N. for MCD and EPR spectroscopy). R.A.L. gratefully acknowledges Florida State University startup funds. D.P.-H. acknowledges the Chilean government through the grant Fondecyt 1180017.

## Author contributions

A.N.G., F.D.W., J.M.S., S.R.S., T.N.P., and T.E.A.-S. conceived, designed, and carried out the synthetic, spectroscopic, electrochemical, and crystallographic experiments with **Bk1**. C.C.-B., M.J.B.-L., and D.P.-H. carried out the theoretical studies. T.J. and J.K.G. conducted the gas-phase experiments. N.J.W. and M.L.N. performed the EPR and MCD experiments. Electrochemical studies of **Ce1** were conducted by N.J.J., A.J.R., and R.A.L. R.E.B. conducted the magnetism studies. D.G.M. conducted IR spectroscopic measurements. All authors discussed and co-wrote the manuscript.

## Competing interests

The authors declare no competing interests.
