## [Peer Review File · Nature Communications]

Creation of an Unexpected Plane of Enhanced Covalency in Cerium(III) and Berkelium(III) Terpyridyl ComplexesREVIEWER COMMENTS

Reviewer #1 (Remarks to the Author):

The authors report on the synthesis of novel complexes of Ce(II) and Bk(III) containing a terpy derivative. This ligand was selected in an attempt to influence the nature of bonding with the metal centre.

The synthesis was well described and the generation of the additional Ce terpy complex was useful in order to provide a reference. Structural characterisation revealed quantitative differences in the bonding interactions of the nitrate ligand lying in the plane of terpy* when compared to those lying above/below the plane, which the authors attribute to the polarization induced by the terpy* ligand. CID Mass spectrometry revealed the binding to terpy* to be weaker than to terpy, consistent with the expected electronic effects.

The authors employ a wide range of computational methods to elucidate the electronic structure of these complexes, and the reason for the choice of methods is not always clear. Attempts to rationalise MCD spectra employed an BP86/STO-TZP model chemistry, although this GGA xc-functional would not be expected to perform particularly well for the task. It appears that the PBE0/def2-TZVP model chemistry was employed to generate initial electronic structures for subsequent post-HF simulations. This choice of model chemistry is more appropriate for DFT studies of these complexes in general, but it begs the questions as to why it not was used elsewhere. Two different sets of CASSCF calculations were performed, using different codes and basis sets, and again it is not clear why more consistency in method couldn't be employed.

EPR reveals interesting features which in Ce1 are beyond my field of expertise to comment on, other than to say that the effects of the terpy* appears to be quite pronounced. CASSCF simulations appear to replicate experimental g-values well, lending credence to the choice of active space which is, as the authors admit, minimal. The authors claim the oblate nature "distributed preferentially along the plane". I assume the authors mean "distributed preferentially in the plane", but even so this assertion is hard to verify from Fig 3 of the ESI, where the visualisations are small and of low quality.

The analysis of chemical bonding is rather thorough, but I am concerned as to it's validity. The choice of active spaces (1,7/12) and (8/7/12) for Ce(III) and Bk(III) are arguably appropriate for ionic systems (although 1-electron active space calculations are effectively HF-calculations that allow for orbital degeneracy and little more) but the authors go to some lengths to demonstrate substantial covalency, which undermines their assertion that this is appropriate. LF-DFT reveals substantial 5p/6p covalent interactions with ligands and, as the authors discuss, visual inspection of the natural orbitals in Figure 4, shows significant ligand contribution to a metal-based 4f/5f-orbital. It is difficult to compare Fig 4 with Supplementary figure 2, but this orbitals appears to be unoccupied in the Ce1. This is indeed strange behaviour if this is a stabilising covalent interaction and should be discussed/clarified. Either way, this implies that the active spaces used may well be inappropriate for bonding studies, with the effect on magnetic properties harder to determine. The authors have clearly considered larger active spaces, as evidenced by the strange CAS(10,10) visualised in supplementary Figure 1 but never discussed. A more appropriate active space would attempt to capture the ligand interactions even if this required RAS approaches.

Reported QAIM reveal apparently extremely large values of rho_BCP which normally be indicative of substantial bond covalency. As a reference, rho_BCP ~ 0.3 a.u. in the highly covalent U-O interactions of uranyl. If the values reported here are correct, they should be discussed in the context of other QAIM studies. To my knowledge, the highest value of rho_BCP in a Ce bond system was ~0.2 a.u. reported by Hayton (doi:10.1021/jacs.6b07932) and typically, values are 0.1 a.u. or less so the reported values appear to be extremely high. This may be due to the units used e^{-1}/A^3 . The literature usually uses a.u., i.e. $\text{e}^{-1}/\text{bohr}^3$ and this may account for the large values.

IQA is an interesting approach to take, but given the lack of correlation included in the bonding interactions here, it is perhaps unsurprising that trends do not reflect the stabilities found in CIDMS. The authors should note that IQA as implemented in AIMAll is based on approximations of the electron pair-density and so energy decomposition will not given components that sum to the total energy (as they should). This, along with the minimal active space, renders this analysis weak, at best.

Overall, while this is an interesting contribution that certainly provides evidence of enhanced bond covalency though judicious choice of ligand set, the rigour of analysis of the bonding of these systems is insufficient to provide a rationalisation of the experimental observations. As such, once the issues above are addressed, I would recommend submission elsewhere.

Minor points:

Page 2. There is confusion regarding state mixing at the bottom of the page. "In the actinide series the splitting is large enough to mix excited states with the ground state" should be reworded. The ground state and the excited state don't mix since by definition they are two different electronic states, but the ground state can be described by the mixing of the lowest energy electronic configuration with "excited" configurations.

Page 3. "...leads to the so-called intermediate coupling regime where no single electronic factor dominates". "Electronic factor" is rather vague and should be clarified"

Supplementary Table 4. Column headings are missing.

Reviewer #2 (Remarks to the Author):

The chemistry of transuranic (TRU) elements is unsurprisingly underdeveloped due to issues associated with the synthesis, storage and safe working with these elements. Here Albrecht-Schönzart and co-workers present a rare structurally characterized Bk complex and an analogous Ce complex of a substituted terpyridyl ligand. The authors find some interesting facets in the structure and bonding of these complexes, and subtle differences between these and a similar Ce terpyridyl complex that lacks such electron-withdrawing substituents. I found the paper very interesting, and well written. The data collected and presented is thought-provoking and should be of wide interest, making it relevant for publication in *Nature Commun*. I have, however, found a significant number of things that need to be addressed in order for it to be complete – there is some discussion that over-extends the data, and there are some gaps in the data and discussion that need to be filled. I trust that the authors will be able to address these issues and I provide some detailed guidance and comments that I hope are of use in improving the manuscript.

The introduction is well-written and puts this work in context. However I do not think that the selected references in the final paragraph are the most appropriate, where primary literature is cited rather than review articles/book chapters. For example, with ref #29 there are multiple examples of symmetry being used to improve actinide SMMs so it would be more representative of the field to cite a review than just one of many papers on this subject, e.g. *Chem. Soc. Rev.*, 2015, 44, 6655 (the context of symmetry in stabilizing low oxidation states for actinides has also been reviewed, e.g. *Organometallics*, 2016, 35, 3088). I am sure that appropriate reviews can be cited for the large binding constant and open coordination site points being made, to replace references 30-38.

Structural characterization: The authors should explicitly mention the intermolecular H-bonding that is present between THF and coordinated water in all three complexes. Whilst the structural difference between the 1Ce/1Bk pair and 2Ce is clear to see there is always an argument to be made for crystal packing forces and intermolecular interactions determining the structures observed in the solid state. If they are able to the authors should address whether solid state structures are maintained in solution or if these are dynamic; perhaps the difference in energy between the two conformations can be calculated on the diamagnetic La analog.

Cyclic voltammetry: The authors state that the electrochemical behavior is irreversible, and then subsequently state that 1Bk is more reversible than 1Ce. I don't find this to make sense; perhaps they mean quasi-reversible instead of irreversible? Irreversible is somewhat an absolute term in this reviewer's mind.

EPR spectroscopy: The authors state that Ce(III) typically exhibits an isotropic EPR spectrum. This is not entirely correct without context – the authors cite a paper containing EPR spectra of an extended solid state material of high site symmetry, which is isotropic, as would be expected. However for molecular complexes it is extremely common for anisotropic spectra to be obtained, wherever anisotropic ligand fields are often present. This discussion needs to be changed; it would be more relevant to cite molecular Ce(III) EPR spectra here over solid state extended lattices.

Moreover, more details are required for the EPR experiment. I would imagine this is an X-band measurement at 10 K or lower on powder samples; these three parameters need to be in the main text and caption as a bare minimum to assess these data. Later in the same section, where calculated g-factors are mentioned, the authors should note that these were calculated by *ab initio* methods, referring the reader to the next section for details.

I disagree with the assertion of the authors that the observation of a magnetic plane and a significant contribution of angular momentum to the g-factor is evidence of covalency in this plane. The g-factors and anisotropy of

lanthanide ligand fields can be calculated using simple electrostatic models and are not related to spatial overlap for Ce. The importance of covalency in determining g-factors in f-block complexes in lanthanides is somewhat overstated throughout this section, where some measure of covalency may only be obtained in an EPR experiment through detailed analysis of ligand superhyperfine interactions. The simple description of the oblate electron distribution of the ground state of Ce(III), its orientation with respect to the ligand field, and the unquenched angular momentum of lanthanides deriving from a lack of interaction with the ligand field, already all accounts for g anisotropy in full for Ln cations (e.g. see Chem. Sci., 2011, 2, 2078 for density plots, and the EPR Spectrum for Complexes of Rare Earth and Actinide Cations chapter in the Electron Paramagnetic Spectroscopy book by P. Bertrand).

Moreover the authors should collect an EPR spectrum of 2Ce for comparison; if differences between the structures of 1Ce and 2Ce are being attributed to electronic effects the authors need to determine the electronic structure of 2Ce in detail (EPR, SQUID and calculations should be performed on 2Ce for full analysis/comparison of magnetic anisotropy).

Calculations: I felt that a qualifier of “predominantly ionic bonding” in these complexes before covalency is discussed in this section is required. This would help the generalist reader keep in their mind that the authors are teasing out smaller contributions of metal-based bonding orbitals to the overall mainly electrostatic-driven bonding situation being analyzed. When the authors raise the relationship of this work with the inverse-trans-influence (ITI), I was somewhat surprised that they did not discuss these results in the context of an earlier report in this same journal (Nat. Commun., 2017, 8, 14137). This earlier work discusses a rare occasion where mid-oxidation state Ln/An complexes are shown to exhibit an ITI, where this is generally only seen for high OS An complexes, so would seem pertinent to consider in the context of these results. I note that one of the py donors of the ligand is trans- to the coordinated water in 1Bk and 1Ce, so why is this example a “plane of covalency” rather than an ITI? I think the authors need to consider this point in their interpretation of the data.

Characterization data: The amount of basic characterization data provided for 1Bk is reasonable, considering its high radioactivity, but the amount of data provided for 1Ce and 2Ce is not enough, as there are no radiological concerns for these compounds. This is especially important for 1Ce, where less data for 1Bk can be collected, as the advantage of making a structurally analogous complex is that the additional data also helps in the confident assignment of 1Bk. My expectation, same as for any new complex reported to be made and isolated, is that 1Ce and 2Ce are scaled up to provide crystalline yields (mass and %), IR, 1H and 13C NMR spectra (though as they are paramagnetic it is likely that no 13C data can be extracted it may be possible to fully assign the 1H spectrum), and either elemental analysis or PXRD, to prove purity of the compounds. I imagine that they have already been prepared on a larger scale than reported here in order to get enough of the compound to get all the characterization data that has been reported. I noted that a La(III) analog was mentioned in the magnetism section of the SI, but there is no mention of this elsewhere – have the Ce and La complexes been fully characterized elsewhere? I did not see citations. If not then all complexes used herein should be fully characterized here, though I note that there may be some solubility issues for solution data.

Supplementary data: The authors need to provide details of all sample preparations performed to obtain characterization data for EPR and SQUID. For EPR spectroscopy they should provide information on tubes used, how powders were ground/solvents dried, spectrometers used, settings employed, etc. Sample preparation and collection strategies for magnetic data is equally of importance to include, especially for consideration of La vs Ce samples, including sample masses and information on diamagnetic corrections etc.

Final supplementary point: As all structures from single crystal XRD have level A and level B alerts in the cifcheck these must be responded to. I imagine that the large number of level A alerts in the Bk structure is an issue of getting the dataset to converge with Bk present, and perhaps this is an intrinsic issue, but I think an explanation is warranted. It would appear that a couple of the alerts in the Ce structures are easily addressed, and the other couple can be left as they are with an explanatory note, but again I would like these responded to.

Minor points:

Page 3 line 4, missing reference.

Page 5, libration > vibration.

All Å symbols missing.

Reviewer #3 (Remarks to the Author):

The contribution by Gaiser et al. describes the synthesis and characterization of a trivalent berkelium complex featuring a substituted terpyridyl ligand. In addition, the isostructural cerium complex and another cerium complex with unsubstituted terpyridine has been synthesized for comparison. The structure of all presented complexes has been determined by single crystal X-ray diffraction. Furthermore, the compounds have been investigated using magnetic (EPR, MCD) and spectroscopic (optical) techniques. Quantum chemical calculations have been performed to compare the bonding properties of the trivalent berkelium to the trivalent cerium complexes.

First of all, I have to admit that working with transcurium elements is limited to only a few institutions worldwide and requires extreme caution and appropriately equipped facilities for handling these extremely radioactive substances. Thus, any structure determination of a berkelium complex helps to broaden our fundamental understanding of these heavy elements and is an important contribution to the field.

That being said I would like to comment on several issues in the manuscript which have to be addressed. My major concern is about the main claim of the manuscript which is an "plane of enhanced covalency" in the reported complexes due to the hyperpolarization of the terpy* ligand.

Two points in this statement have to be discussed more closely:

First, the "plane" of covalency seems to only affect the bound water molecule, as the nitrate which is located in this "plane" neither possesses shorter M-O bond distances than the axial nitrates nor shows significant differences to the axial nitrates in the quantum chemical bond metrics as shown in Figure 5. The authors have to explain this.

Second, the term "enhanced" covalency has to be explained more clearly. The results from QTAIM and IQA indicate a similar degree of covalency for both Bk and Ce complexes which is rather unexpected, though. However, the difference in Ce-OH₂ bond length between both cerium complexes cannot be solely attributed to the effect of the 4-Nitrophenyl substituent if there is no comparison of these two complexes on a quantum chemical basis. Crystallographic packing effects may influence the measured distances. So, the authors have to clarify the "enhancement" of covalency in comparison to a specified reference system. How do the values compare to the investigations of the isostructural lanthanide and americium complexes (see Inorg. Chem. 2018, 57, 12969–12975)? Is cerium the odd one out of the lanthanide series or is a similar "enhancement" also observed for the other lanthanides?

If these major concerns are appropriately addressed by the authors, I am willing to accept publication in Nature Communications.

In addition to these major points some other issues need explanation:

- **Synthesis:** Although the major point of this manuscript is the difficult synthesis of the berkelium complex also the isostructural cerium complexes should be characterized according to standard solid-state analytical techniques like elemental analysis or IR spectroscopy to confirm the bulk purity. Also the yield of the syntheses should be mentioned in the supporting information.
- **SC-XRD:** Something is odd with the refinement of the berkelium crystal structure. According to the checkcif many alerts of A and B level are present. The authors should at least comment on each alert also in the cerium structures to warrant their discussions based on the determined interatomic distances in the manuscript. Furthermore, additional details about the refinement process including the final R-values and the bond distances (as indicated on p.6 I.2 in the manuscript) are missing in the Supporting Material.
- **Optical spectroscopy:** Although f-f transitions of Bk^{III} are present in the spectrum, charge transfer bands due to Bk^{IV} below 300 nm cannot be ruled out. Thus, the statement concerning the purity of Bk's oxidation state on page 7 has to be softened.
- **Gas Phase studies:** To me this very conclusive investigation in the gas phase is not properly connected to the other results presented in this manuscript. How do the observations on the higher binding strength to Ce than Eu and to the unsubstituted terpy ligand correspond to the remaining analytical techniques? It is clear that it may not be possible to measure the berkelium samples with the similar technique but the authors have to indicate how these results help to conclude that berkelium and cerium show a plane of enhanced covalency for the substituted terpy* ligand. The results obtained from SC-XRD indicate that the covalent character and hence the binding strength is less pronounced in the unsubstituted terpy ligand. Isn't this contradictory to the results obtained in the gas phase studies? Furthermore, the comparison to europium should be included in the calculations as some

QTAIM results have already been published by the same group (Inorg. Chem. 2018, 57, 12969–12975).

- MCD spectroscopy: Again, the benefit of this analytical technique to the purpose of the manuscript has to be explained in more detail.

Minor issues:

- p.3 l.4: correct citation is missing

- p.10 l.7: nitro group has to be replaced by nitrate

- Caption of figure 1: The authors should indicate which atoms they used to define the plane. I assume the three nitrogen atoms of the terpy ligand, the oxygen atom of the water and the nitrogen atom of the nitrate have been used. The term “the terpyridine ring” is misleading and not correct.

- Figure 5: The authors should change the order of the bars (Bk top, Ce bottom) in b3, c2 and c3 to have the same order in all figures. Furthermore, the unit of the different energies of the IQA analysis have to be indicated.

RESPONSE TO REFEREES

Reviewer #1 (Remarks to the Author):

The authors employ a wide range of computational methods to elucidate the electronic structure of these complexes, and the reason for the choice of methods is not always clear. Attempts to rationalise MCD spectra employed an BP86/STO-TZP model chemistry, although this GGA xc-functional would not be expected to perform particularly well for the task. It appears that the PBE0/def2-TZVP model chemistry was employed to generate initial electronic structures for subsequent post-HF simulations. This choice of model chemistry is more appropriate for DFT studies of these complexes in general, but it begs the questions as to why it not was used elsewhere. Two different sets of CASSCF calculations were performed, using different codes and basis sets, and again it is not clear why more consistency in method couldn't be employed.

We agree with the reviewer in that more clarification about the computational details were needed. The section was rewritten to clarify the reviewer's concerns. Though, we respectfully disagree with the reviewer in the use of a GGA functional for assigning the electronic transitions of the MCD spectra. We rely on the system dependency of DFT functionals and the BP86/STO-TZP is the level of theory that shown the best match with the experimental spectra. With respect to the use of two codes, we apologize for the mistake. The results correspond to the wavefunctions produced with MOLCAS, we originally started with ORCA but state-specific wavefunctions were unable to converge in ORCA and in the process of writing, we did not update the details.

EPR reveals interesting features which in CeI are beyond my field of expertise to comment on, other than to say that the effects of the terpy appears to be quite pronounced. CASSCF simulations appear to replicate experimental g-values well, lending credence to the choice of active space which is, as the authors admit, minimal. The authors claim the oblate nature "distributed preferentially along the plane". I assume the authors mean "distributed preferentially in the plane", but even so this assertion is hard to verify from Fig 3 of the ESI, where the visualisations are small and of low quality.*

Supplementary Fig. 3 was replaced by a higher resolution picture. The text was fixed according to the reviewer's suggestions. We would like to clarify that the distribution in the plane corresponds to only KD1.

The analysis of chemical bonding is rather thorough, but I am concerned as to it's validity. The choice of active spaces (1,7/12) and (8/7/12) for Ce(III) and Bk(III) are arguably appropriate for ionic systems (although 1-electron active space calculations are effectively HF-calculations that allow for orbital degeneracy and little more) but the authors go to some lengths to demonstrate substantial covalency, which undermines their assertion that this is appropriate. LF-DFT reveals substantial 5p/6p covalent interactions with ligands and, as the authors discuss, visual inspection of the natural orbitals in Figure 4, shows significant ligand contribution to a metal-based 4f/5f-orbital. It is difficult to compare Fig 4 with Supplementary figure 2, but this orbitals appears to

be unoccupied in the Ce1. This is indeed strange behaviour if this is a stabilising covalent interaction and should be discussed/clarified. Either way, this implies that the active spaces used may well be inappropriate for bonding studies, with the effect on magnetic properties harder to determine. The authors have clearly considered larger active spaces, as evidenced by the strange CAS(10,10) visualised in supplementary Figure 1 but never discussed. A more appropriate active space would attempt to capture the ligand interactions even if this required RAS approaches.

We understand the reviewer's concern, and we agree in the sense that the active spaces should be larger than the ones we considered. However, finding a larger active space but keeping the balance between the orbitals (bonding-antibonding) is impractical. It would be ideal to find the antibonding counterpart of the f-bonding orbital but it was not possible. However, even in the best case scenario it is known that post-HF wavefunctions over localize the density while DFT over delocalizes. If these wavefunctions display a certain degree of covalency, at least we know that we approach covalency from the underestimation of it, which is more prudent. DFT on the other hand may provide a better density, but the amount of dynamic correlation recovered is unknown.

The inclusion of the seven f orbitals + the unoccupied d shell (unbalanced for not including the bonding counterparts) were possible and no changes were observed in the g-factors. When ligand-based bonding or antibonding orbitals with respect to the f- and d-orbitals were attempted to be included to the active space the complexity of the bonding pattern within the ligands made impossible to find the metal counterparts. Therefore, the complexity of these systems prevents us to expand the active spaces but in order to clarify this; we added a paragraph discussing these difficulties and clarifying that larger active spaces should be considered when possible to provide a more accurate description of the bonding.

The discussion of semi-core 5p/6p covalency is a whole different problem where impractical active spaces should be considered. That is why the LFDFT approach was used for this purpose.

With respect to the orbitals shown in Supplementary Fig. 2, we understand the concern of the reviewer but this set of orbitals corresponds to state-average orbitals and not state-specific, which is what we used for bonding analysis. This has been clarified in the figure captions. The state-specific orbitals in Ce1 show the "bonding" f-orbital that is missing in the state-average set.

Reported QTAIM reveal apparently extremely large values of ρ_{BCP} which normally be indicative of substantial bond covalency. As a reference, $\rho_{BCP} \sim 0.3$ a.u. in the highly covalent U-O interactions of uranyl. If the values reported here are correct, they should be discussed in the context of other QTAIM studies. To my knowledge, the highest value of ρ_{BCP} in a Ce bond system was ~ 0.2 a.u. reported by Hayton (doi:10.1021/jacs.6b07932) and typically, values are 0.1 a.u. or less so the reported values appear to be extremely high. This may be due to the units used $|e|/A^3$. The literature usually uses a.u., i.e. $|e|/\text{bohr}^3$ and this may account for the large values.

Our QTAIM values might seem larger to the reviewer but it is only due to units we are using. Some literature uses the units we adopted for this paper due to the atomic units are smaller in order and so, more difficult to compare. Therefore, we do not think necessary to compare against uranyl because the QTAIM metrics we report here are not as covalent as in uranyl. If the reviewer is

curious about the unit difference, the conversion factor increases our electron density values roughly an order of magnitude with respect to atomic units.

IQA is an interesting approach to take, but given the lack of correlation included in the bonding interactions here, it is perhaps unsurprising that trends do not reflect the stabilities found in CIDMS. The authors should note that IQA as implemented in AIMAll is based on approximations of the electron pair-density and so energy decomposition will not given components that sum to the total energy (as they should). This, along with the minimal active space, renders this analysis weak, at best.

We thank the reviewer for this comment and agree that IQA approximate the 2-electron density matrix from the 1EDM introducing uncertainty in the total energies. We ran HF calculations, where the 2EDM is calculated exactly to confirm that the trends are correct. Along these lines, we still think these results are relevant for the manuscript. With respect to the correlation, the reviewer is right, but in order to recover the correlation necessary to capture the real nature of the bond, an active space of more than 20 orbitals should be included, which is impractical. Otherwise the active space would be unbalanced and the results even less reliable.

Regarding the correlation with CIDMS, we do not understand the reviewer since computations were performed only on BkTpy* and CeTpy*, while CIDMS was performed on Ce and Eu Tpy and Tpy* systems. The only common structure is CeTpy*, therefore no correlation can be made.

However, we think that modifications needed to be made to the text in order to clarify the reviewer's concerns. Thus, the IQA section in the main text was reworded and focused only on the exchange (covalent) term, which is where the strength of the interaction resides according to M.A. Pendas. The Computational details were also modified accordingly.

Minor points:

Page 2. There is confusion regarding state mixing t the bottom of the page. "In the actinide series the splitting is large enough to mix excited states with the ground state" should be reworded. The ground state and the excited state don't mix since by definition they are two different electronic states, but the ground state can be described by the mixing of the lowest energy electronic configuration with "excited" configurations.

The sentence has been reworded.

Page 3. "...leads to the so-called intermediate coupling regime where no single electronic factor dominates". "Electronic factor" is rather vague and should be clarified"

The sentence has been reworded.

Supplementary Table 4. Column headings are missing.

The table has been fixed.

Reviewer #2 (Remarks to the Author):

The introduction is well-written and puts this work in context. However I do not think that the selected references in the final paragraph are the most appropriate, where primary literature is cited rather than review articles/book chapters. For example, with ref #29 there are multiple examples of symmetry being used to improve actinide SMMs so it would be more representative of the field to cite a review than just one of many papers on this subject, e.g. Chem. Soc. Rev., 2015, 44, 6655 (the context of symmetry in stabilizing low oxidation states for actinides has also been reviewed, e.g. Organometallics, 2016, 35, 3088). I am sure that appropriate reviews can be cited for the large binding constant and open coordination site points being made, to replace references 30-38.

We agree with this suggestion in part. Given that entire area of research is being cited, referencing a review is certainly warranted, and we have done this as suggested for reference #29. However, referencing the primary literature is paramount, and we have retained the above references as well.

Structural characterization: The authors should explicitly mention the intermolecular H-bonding that is present between THF and coordinated water in all three complexes. Whilst the structural difference between the 1Ce/1Bk pair and 2Ce is clear to see there is always an argument to be made for crystal packing forces and intermolecular interactions determining the structures observed in the solid state. If they are able to the authors should address whether solid state structures are maintained in solution or if these are dynamic; perhaps the difference in energy between the two conformations can be calculated on the diamagnetic La analog.

We kindly thank the reviewer for pointing out the crystal packing argument and have added a sentence accordingly into the manuscript in addition to one noting the THF hydrogen bonding.

Cyclic voltammetry: The authors state that the electrochemical behavior is irreversible, and then subsequently state that 1Bk is more reversible than 1Ce. I don't find this to make sense; perhaps they mean quasi-reversible instead of irreversible? Irreversible is somewhat an absolute term in this reviewer's mind.

The authors agree that the term irreversible is absolute and have made changes to the text to describe the system as quasi-reversible, rather than irreversible.

EPR spectroscopy: The authors state that Ce(III) typically exhibits an isotropic EPR spectrum. This is not entirely correct without context – the authors cite a paper containing EPR spectra of an extended solid state material of high site symmetry, which is isotropic, as would be expected. However for molecular complexes it is extremely common for anisotropic spectra to be obtained, wherever anisotropic ligand fields are often present. This discussion needs to be changed; it would be more relevant to cite molecular Ce(III) EPR spectra here over solid state extended lattices.

The sentence has been reworded accordingly.

Moreover, more details are required for the EPR experiment. I would imagine this is an X-band measurement at 10 K or lower on powder samples; these three parameters need to be in the main text and caption as a bare minimum to assess these data. Later in the same section, where calculated g-factors are mentioned, the authors should note that these were calculated by ab initio methods, referring the reader to the next section for details.

We have added an informational paragraph to the experimental section of the SI and have reworded the sentence in the main text.

I disagree with the assertion of the authors that the observation of a magnetic plane and a significant contribution of angular momentum to the g-factor is evidence of covalency in this plane. The g-factors and anisotropy of lanthanide ligand fields can be calculated using simple electrostatic models and are not related to spatial overlap for Ce. The importance of covalency in determining g-factors in f-block complexes in lanthanides is somewhat overstated throughout this section, where some measure of covalency may only be obtained in an EPR experiment through detailed analysis of ligand superhyperfine interactions. The simple description of the oblate electron distribution of the ground state of Ce(III), its orientation with respect to the ligand field, and the unquenched angular momentum of lanthanides deriving from a lack of interaction with the ligand field, already all accounts for g anisotropy in full for Ln cations (e.g. see Chem. Sci., 2011, 2, 2078 for density plots, and the EPR Spectrum for Complexes of Rare Earth and Actinide Cations chapter in the Electron Paramagnetic Spectroscopy book by P. Bertrand).

While the reviewer is correct that superhyperfine interactions can provide more detailed insight into the covalency in these systems than g-values, the deviation of the g-values from spin-only values in f-block systems can also give insight into covalency (the orbital angular momentum is further reduced beyond the ligand field due to covalency as the unpaired electron is not in a pure f-orbital). This was highlighted in actinides in the series reported by Lukens and Hayton in 2013 (*J. Am. Chem. Soc.* 2013, 135, 29, 10742–10754). While it is true that the effects observed for lanthanides would be smaller than that of actinides, the same fundamental electronic effects stand. However, they are also correct that the anisotropy can arise from the other effects. Just that the absolute g-values can be affected by covalency as stated above.

Moreover the authors should collect an EPR spectrum of 2Ce for comparison; if differences between the structures of 1Ce and 2Ce are being attributed to electronic effects the authors need to determine the electronic structure of 2Ce in detail (EPR, SQUID and calculations should be performed on 2Ce for full analysis/comparison of magnetic anisotropy)

We thank the reviewer for this point, but the purpose of presenting Ce₂ is solely for structural comparison. The focus of the manuscript is on Bk₁ and Ce₁. Ce₂ is included solely to point out the difference in water bond length. Ce₂ is actually a compound previously known from the literature.

Calculations: I felt that a qualifier of “predominantly ionic bonding” in these complexes before

covalency is discussed in this section is required. This would help the generalist reader keep in their mind that the authors are teasing out smaller contributions of metal-based bonding orbitals to the overall mainly electrostatic-driven bonding situation being analyzed. When the authors raise the relationship of this work with the inverse-trans-influence (ITI), I was somewhat surprised that they did not discuss these results in the context of an earlier report in this same journal (Nat. Commun., 2017, 8, 14137). This earlier work discusses a rare occasion where mid-oxidation state Ln/An complexes are shown to exhibit an ITI, where this is generally only seen for high OS An complexes, so would seem pertinent to consider in the context of these results. I note that one of the py donors of the ligand is trans- to the coordinated water in 1Bk and 1Ce, so why is this example a “plane of covalency” rather than an ITI?

We thank the reviewer for pointing this out. We added the clarification regarding the nature of the bond in the f-block as predominantly ionic with a certain degree of covalency. With respect to the ITI, we deleted the final sentence of that discussion. It was already briefly discussed in the text where we also added the reference suggested by the reviewer.

Characterization data: The amount of basic characterization data provided for 1Bk is reasonable, considering its high radioactivity, but the amount of data provided for 1Ce and 2Ce is not enough, as there are no radiological concerns for these compounds. This is especially important for 1Ce, where less data for 1Bk can be collected, as the advantage of making a structurally analogous complex is that the additional data also helps in the confident assignment of 1Bk. My expectation, same as for any new complex reported to be made and isolated, is that 1Ce and 2Ce are scaled up to provide crystalline yields (mass and %), IR, 1H and 13C NMR spectra (though as they are paramagnetic it is likely that no 13C data can be extracted it may be possible to fully assign the 1H spectrum), and either elemental analysis or PXRD, to prove purity of the compounds. I imagine that they have already been prepared on a larger scale than reported here in order to get enough of the compound to get all the characterization data that has been reported. I noted that a La(III) analog was mentioned in the magnetism section of the SI, but there is no mention of this elsewhere – have the Ce and La complexes been fully characterized elsewhere? I did not see citations. If not then all complexes used herein should be fully characterized here, though I note that there may be some solubility issues for solution data.

We determined crystalline yields, elemental analysis and PXRD for both Ce1 and Ce2 and have added the data into the ESI. The La(III) analog has been characterized in the IC paper by this group cited in the manuscript on page 5 (#43). We included magnetism data for the La analog in this paper solely to provide a sample baseline for the Ce data.

Supplementary data: The authors need to provide details of all sample preparations performed to obtain characterization data for EPR and SQUID. For EPR spectroscopy they should provide information on tubes used, how powders were ground/solvents dried, spectrometers used, settings employed, etc. Sample preparation and collection strategies for magnetic data is equally of importance to include, especially for consideration of La vs Ce samples, including sample masses and information on diamagnetic corrections etc.

We have updated the SI to include more details on EPR and SQUID measurements.

Final supplementary point: As all structures from single crystal XRD have level A and level B alerts in the cifcheck these must be responded to. I imagine that the large number of level A alerts in the Bk structure is an issue of getting the dataset to converge with Bk present, and perhaps this is an intrinsic issue, but I think an explanation is warranted. It would appear that a couple of the alerts in the Ce structures are easily addressed, and the other couple can be left as they are with an explanatory note, but again I would like these responded to.

All of the level A alerts for the Bk structure are due to checkcif not recognizing berkelium. The majority of the level B alerts also come up for this reason. The long O-H bond length level B alert is due to the water hydrogen atoms not being automatically assigned, but instead labeled from the accumulation of electron density. The long bond length is with the hydrogen atom from the water molecule that is hydrogen bonding with the oxygen atom on the THF molecule. The short N—O interactions are roughly 2.85 Å between the outer most nitrogen atoms on the terpyridine rings and the closest oxygen atom on a nitrate molecule. The length is comparable to those within the Ce analog which are 2.9 and 2.95 Å. This alert can be explained through the contraction that occurs when data is collected at 28 K.

Minor points:

Page 3 line 4, missing reference.

Reference has been fixed.

Page 5, libration > vibration.

The sentence has been reworded.

All Å symbols missing.

All Å symbols have been checked.

Reviewer #3 (Remarks to the Author):

My major concern is about the main claim of the manuscript which is an “plane of enhanced covalency” in the reported complexes due to the hyperpolarization of the terpy ligand. Two points in this statement have to be discussed more closely:*

First, the “plane” of covalency seems to only affect the bound water molecule, as the nitrate which

is located in this “plane” neither possesses shorter M-O bond distances than the axial nitrates nor shows significant differences to the axial nitrates in the quantum chemical bond metrics as shown in Figure 5. The authors have to explain this.

We have significantly altered this paragraph to improve clarity. One nitrate anion that binds the M(III) centers is bisected by the plane of polarization. The two N-O bonds of this nitrate anion show differential behavior with respect to the two nitrate anions that are above and below the polarization plane. This is discussed in the third paragraph under structural characterization.

*Second, the term “enhanced” covalency has to be explained more clearly. The results from QTAIM and IQA indicate a similar degree of covalency for both Bk and Ce complexes which is rather unexpected, though. However, the difference in Ce-OH₂ bond length between both cerium complexes cannot be solely attributed to the effect of the 4-Nitrophenyl substituent if there is no comparison of these two complexes on a quantum chemical basis. Crystallographic packing effects may influence the measured distances. So, the authors have to clarify the “enhancement” of covalency in comparison to a specified reference system. How do the values compare to the investigations of the isostructural lanthanide and americium complexes (see *Inorg. Chem.* 2018, 57, 12969–12975)? Is cerium the odd one out of the lanthanide series or is a similar “enhancement” also observed for the other lanthanides?*

Sentences have been added to accommodate the possibility of crystallographic packing effects. The isostructural lanthanide and americium complexes do not exhibit the same enhancement. This can be seen in the SI of the *Inorg. Chem.* paper.

In addition to these major points some other issues need explanation:

- *Synthesis: standard solid-state analytical techniques like elemental analysis or IR spectroscopy to confirm the bulk purity. Also the yield of the syntheses should be mentioned in the supporting information.*

pXRD was performed to confirm the bulk purity and added to the SI.

- *SC-XRD: berkelium checkcif many alerts of A and B level are present. the cerium structures warrant their discussions additional details about the refinement process including the final R-values and the bond distances (as indicated on p.6 l.2 in the manuscript) are missing in the Supporting Material.*

All of the level A alerts for the Bk structure are due to checkcif not recognizing berkelium. The majority of the level B alerts also come up for this reason. The long O-H bond length level B alert is due to the water hydrogen atoms not being automatically assigned, but instead labeled from the accumulation of electron density. The long bond length is with the hydrogen atom from the water molecule that is hydrogen bonding with the oxygen atom on the THF molecule. The short N—O interactions are roughly 2.85 Å between the outer most nitrogen atoms on the terpyridine rings and

the closest oxygen atom on a nitrate molecule. The length is comparable to those within the Ce analog which are 2.9 and 2.95 Å. This alert can be explained through the contraction that occurs when data is collected at 28 K.

- *Optical spectroscopy: Although f-f transitions of Bk(III) are present in the spectrum, charge transfer bands due to Bk(IV) below 300 nm cannot be ruled out. Thus, the statement concerning the purity of Bk's oxidation state on page 7 has to be softened.*

Photoluminescence of the sample was measured and no luminescence that could be attributed to Bk(IV) was observed. Additionally, the Bk1 bond lengths correlate well with other Bk(III) structures, but not with Bk(IV). If there is any Bk(IV) present, it cannot be detected. Synthesis of Ce(IV) analog was attempted multiple times and always resulted in isolation of the Ce(III) (Ce1) compound.

- *Gas Phase studies: To me this very conclusive investigation in the gas phase is not properly connected to the other results presented in this manuscript. How do the observations on the higher binding strength to Ce than Eu and to the unsubstituted terpy ligand correspond to the remaining analytical techniques? It is clear that it may not be possible to measure the berkelium samples with the similar technique but the authors have to indicate how these results help to conclude that berkelium and cerium show a plane of enhanced covalency for the substituted terpy* ligand. The results obtained from SC-XRD indicate that the covalent character and hence the binding strength is less pronounced in the unsubstituted terpy ligand. Isn't this contradictory to the results obtained in the gas phase studies? Furthermore, the comparison to europium should be included in the calculations as some QTAIM results have already been published by the same group (Inorg. Chem. 2018, 57, 12969–12975).*

The gas phase studies were not performed to support the plane of covalency, but to provide further insight on how differential Ce can bind to terpy and terpy* with respect to a Ln analog such as Eu. The covalent character obtained from solid-state structures is a result of more interactions than those found in the gas phase. In fact, the structures in the gas phase differ from the solid-state complexes due to the fragmentation process, and therefore they are not comparable. Thus, the only interaction that is “comparable” would be the metal- terpy and terpy*.

A comparison with the QTAIM metrics reported previously has been included in the main text.

- *MCD spectroscopy: Again, the benefit of this analytical technique to the purpose of the manuscript has to be explained in more detail.*

We have edited the text to reflect the importance of the MCD measurements.

Minor issues:

- p.3 l.4: correct citation is missing

Citation has been fixed.

- p.10 l.7: nitro group has to be replaced by nitrate

Sentence has been reworded.

- *Caption of figure 1: The authors should indicate which atoms they used to define the plane. I assume the three nitrogen atoms of the terpy ligand, the oxygen atom of the water and the nitrogen atom of the nitrate have been used. The term “the terpyridine ring” is misleading and not correct.*

Terpyridine ring has been changed to “terpyridine derivative”. The atoms used to optimize the plane are now explicitly included in the figure caption.

- *Figure 5: The authors should change the order of the bars (Bk top, Ce bottom) in b3, c2 and c3 to have the same order in all figures. Furthermore, the unit of the different energies of the IQA analysis have to be indicated.*

The figure has been replaced accordingly, and units have been added to the figure caption.

REVIEWER COMMENTS

Reviewer #1 (Remarks to the Author):

The reviewer appreciates the efforts made to address the concerns raised with regard to the original manuscript. While there is inevitably room for improvement when it comes to simulation data in particular, I am happy that the approach presented is justifiable and so can now recommend publication

Reviewer #2 (Remarks to the Author):

I thank the authors for their response and their changes to the article. Some of the concerns raised have not been fully addressed, and I would expect the following points to be addressed in full before publication:

1. The authors appear to have misunderstood the point made regarding the referencing in the introduction, so it is clarified here. New refs#33-39 are supposed to represent literature on “open sites around actinides that lead to unique reactivities”, and the authors state that “referencing the primary literature is paramount” in their rebuttal. However, with hundreds of articles to choose from, why these seven references? There must be some bias introduced in such a procedure if it is not done randomly, and there are currently 1-2 references each to some of the most well-known corresponding authors in the field. The choice of these articles out of these authors’ portfolios do not always seem to be the most logical choices either to match the sentence in the introduction. To this referee this approach doesn’t correlate to appropriate citation of the literature and I request that the authors address this point by citing reviews instead – this is certainly more apt than a huge list of primary literature or the handful of references provided here.

2. The authors state in their response that the SI now contains crystalline yields, elemental analysis and PXRD for Ce1 and Ce2 to address the paucity of data comment. I was unable to find crystalline yields or elemental analysis results for either complex when searching through the new SI, and could only see PXRD for Ce1. Not only should these data be added, the authors should also perform the basic ¹H NMR and IR spectroscopy measurements requested. As stated previously a procedure for 1Ce on a preparative scale should be included, not just the small scale to match that of 1Bk.

3. The authors have not yet addressed the confusion caused to this referee not being sure what complexes are already known and what are new – other readers will be equally confused. The authors should clearly state which complexes were synthesized according to literature procedures in a sentence with citations. Note that reference #44 for the La analog is currently missing page numbers, but this referee still doesn’t know where to find the previous synthesis of 2Ce after reading the article a second time.

4. The authors have responded to the level A and B alerts in the cifchecks of the Bk and Ce complexes but have not provided the updated cifs and cifchecks in the set of revised files. I would need to see that these responses have been incorporated in these files; there are no CCDC/CSD numbers provided for me to access these independently so can the authors please provide them.

5. Upon addressing referee comments the authors have toned down some claims from the original submission, noting that both the XRD and EPR data are not unequivocal experimental proof of a “plane of covalency.” The remaining evidence is from calculations, which as the authors acknowledge in their response to referee#1, are not perfect due to compromises required to make these calculations feasible. Given the paucity of categorical evidence the authors should consider a change of manuscript title to one that is less absolute and more reflective of the language in the abstract and conclusion of the terpy* ligand inducing a highly anisotropic environment.

Reviewer #3 (Remarks to the Author):

The authors have revised their manuscript according to the reviewer's suggestions. They have properly addressed all concerns raised by the reviewers. Furthermore, the manuscript has definitely improved by the addition of explanations to each analytical method and the connection between these.

I also acknowledge the efforts of the authors to characterize the bulk material of Ce1 by PXRD. Although the bulk characterization of the compound Ce2 is still missing in the current manuscript I am willing to accept it for publication in Nature Communications as it is.

Reviewer #1 (Remarks to the Author):

The reviewer appreciates the efforts made to address the concerns raised with regard to the original manuscript. While there is inevitably room for improvement when it comes to simulation data in particular, I am happy that the approach presented is justifiable and so can now recommend publication

Response: We appreciate the Reviewers support of this publication.

Reviewer #2 (Remarks to the Author):

I thank the authors for their response and their changes to the article. Some of the concerns raised have not been fully addressed, and I would expect the following points to be addressed in full before publication:

1. The authors appear to have misunderstood the point made regarding the referencing in the introduction, so it is clarified here. New refs#33-39 are supposed to represent literature on “open sites around actinides that lead to unique reactivities”, and the authors state that “referencing the primary literature is paramount” in their rebuttal. However, with hundreds of articles to choose from, why these seven references? There must be some bias introduced in such a procedure if it is not done randomly, and there are currently 1-2 references each to some of the most well-known corresponding authors in the field. The choice of these articles out of these authors’ portfolios do not always seem to be the most logical choices either to match the sentence in the introduction. To this referee this approach doesn’t correlate to appropriate citation of the literature and I request that the authors address this point by citing reviews instead – this is certainly more apt than a huge list of primary literature or the handful of references provided here.

Response: With all due respect, we have a different philosophy when it comes to referencing than the reviewer. We respect the reviewer’s, and they should respect ours. Reviews are not comprehensive and become outdated with time. Furthermore, we prefer referencing the primary literature when possible and have done this for decades as do many others in the field. We compromised with the reviewer and added a review in one section, but we are not changing this further. Our choices of references are not “random,” but rather represent what we consider to be good examples. Given the total number of reference restrictions for this journal, we cannot expand this further.

2. The authors state in their response that the SI now contains crystalline yields, elemental analysis and PXRD for Ce1 and Ce2 to address the paucity of data comment. I was unable to find crystalline yields or elemental analysis results for either complex when searching through the new SI, and could only see PXRD for Ce1. Not only should these data be added, the authors should also perform the basic ¹H NMR and IR spectroscopy measurements requested. As stated previously a procedure for 1Ce on a preparative scale should be included, not just the small scale to match that of 1Bk.

Response: The experimental yield has been added to the SI. Elemental analysis and IR spectroscopy were not performed because the purity of **Ce1** was confirmed with PXRD. The IR spectra do not provide meaningful information. We also do not have IR capabilities in our lab and adding new instrumentation to nuclear facilities takes years. In this case, it would not be a good investment of resources, or we would have done it years ago. Unfortunately, **Ce1** is not sufficiently

soluble in any available deuterated solvent to allow for collecting ^1H NMR spectra. This not surprising – these donor-acceptor polyaromatics are well-known to be sparingly soluble in most solvents. The only solution technique that is sensitive enough at these concentrations (in THF) is cyclic voltammetry, which we did conduct. The small-scale synthesis (5 mg) is the same as was done for a larger scale (50 mg) of **Ce1**, so an identical synthetic method was not added to the SI. **Ce2** pXRD is not reported because it degrades during grinding (probably because of lattice solvent loss). We explored this carefully and the PXRD patterns evolve as the sample is ground. The preferred orientation of the crystals makes the pattern measured from an unground sample uninformative. Moreover, the purpose of reporting **Ce2** is for the low-temperature bond length comparison of a water molecule that falls outside the plane of enhanced covalent character exhibited in **Ce1** and for the gas-phase BDI measurements. Neither of these data would be affected by the presence of an impurity. We have not discussed the bulk properties of **Ce2** in the manuscript, and thus, the crystal structure is enough for this benchmark compound.

3. The authors have not yet addressed the confusion caused to this referee not being sure what complexes are already known and what are new – other readers will be equally confused. The authors should clearly state which complexes were synthesized according to literature procedures in a sentence with citations. Note that reference #44 for the La analog is currently missing page numbers, but this referee still doesn't know where to find the previous synthesis of 2Ce after reading the article a second time.

Response: This was a very helpful observation that we confounded further with our last response. Both compounds are new. **Ce2** is not reported in the CCDC. It appears that this is the first time this structure has been reported. Other lanthanide analogs of **Ce2** are in literature. However, they are synthesized using a different method than we employed. Reference #44 has been updated to include page numbers.

4. The authors have responded to the level A and B alerts in the cifchecks of the Bk and Ce complexes but have not provided the updated cifs and cifchecks in the set of revised files. I would need to see that these responses have been incorporated in these files; there are no CCDC/CSD numbers provided for me to access these independently so can the authors please provide them.

Response: We thank the reviewer for the comment. 2050447, 1857536, and 2050448 are the CCDC numbers for **Bk1**, **Ce1**, and **Ce2** respectively. These codes have been added to the manuscript under Accession Codes. Please find the cifs and check cifs attached.

5. Upon addressing referee comments the authors have toned down some claims from the original submission, noting that both the XRD and EPR data are not unequivocal experimental proof of a “plane of covalency.” The remaining evidence is from calculations, which as the authors acknowledge in their response to referee#1, are not perfect due to compromises required to make these calculations feasible. Given the paucity of categorical evidence the authors should consider a change of manuscript title to one that is less absolute and more reflective of the language in the abstract and conclusion of the terpy* ligand inducing a highly anisotropic environment.

Response: The experimental and computation results all point to the same conclusion when taken together. We are declining changing the title of the manuscript.

Reviewer #3 (Remarks to the Author):

The authors have revised their manuscript according to the reviewer's suggestions. They have properly addressed all concerns raised by the reviewers. Furthermore, the manuscript has definitely improved by the addition of explanations to each analytical method and the connection between these.

I also acknowledge the efforts of the authors to characterize the bulk material of Ce1 by PXRD. Although the bulk characterization of the compound Ce2 is still missing in the current manuscript I am willing to accept it for publication in Nature Communications as it is.

Response: We thank this reviewer for their support.

REVIEWER COMMENTS

Reviewer #2 (Remarks to the Author):

I thank the authors for their second response, and will not debate subjective points further.

In the first rebuttal the authors claimed to have collected elemental analysis and PXRD data for both Ce1 and Ce2 as requested. When I raised that I could not locate these data in the resubmission, the authors now respond that they have not collected any elemental analysis data for either complex, and only provide the PXRD of Ce1. In their first rebuttal the authors stated that complex Ce2 was a known compound already in the literature, and in their second response they now recognise that it is a new compound reported here for the first time. This inconsistency is confusing.

The authors state that they do not need elemental analysis or IR spectroscopy data, and provide a variety of reasons for the latter, including difficulties in collecting an IR spectrum due to a lack of a spectrometer in the nuclear facility and a dissatisfaction with the technique not providing meaningful information. The lack of ¹H NMR data for both complexes is attributed to poor solubility in the solvents investigated. No response is given for the lack of elemental analysis data for Ce2. The lack of PXRD data for Ce2 is attributed to sample degradation upon grinding, and in the authors' opinion is mitigated by the fact that only a structural comparison is needed.

Responding to the above, I direct the authors to the journal guidelines:

<https://www.nature.com/ncomms/submit/chemical-characterisation>

"Yield and evidence of sample purity is required for each isolated compound."

Whilst exceptions can (and should) be made for radiochemicals, these guidelines should apply to Ce1 and Ce2, and it is an expectation of the journal that the authors provide these data. I accept that a paucity of NMR data is common for paramagnetic complexes, and the solubility issues are noted. ¹H NMR spectroscopy should be attempted in a solvent that several mg of the complexes can dissolve in; can the authors please attempt a logical reasonable selection of solvents according to their judgement (e.g. d₂-DCM, d₅-chlorobenzene, d₈-THF, d₅-pyridine, d₃-MeCN). I note that the authors have stated that the complexes did not dissolve in any available NMR solvent, but it would be useful to know what was available and what has been attempted.

In the absence of any solution NMR data a common basic requirement for reporting paramagnetic complexes in this referees experience is to provide elemental analyses and IR spectra. These are typically "cheaper" measurements to perform over other standard techniques, e.g. EPR spectroscopy and magnetometry, so they were suggested for perceived simplicity. To confirm the bulk purity of Ce2 the authors should provide elemental analysis results or other convincing evidence. The IR spectra were requested to mitigate the lack of NMR data provided; whilst accepting that these data are often not as useful, they can provide a fingerprint of molecular vibrational modes. If there are issues with collecting IR spectra in a nuclear facility is there any capacity to collect the IR spectra of Ce1 and Ce2 outside of this facility? These data should be provided if possible. This is a compromise from the initial request to collect SQUID and EPR data for Ce2, taking into account the authors' response.

The authors have also still not included the mass yields of Ce1 and Ce2 products. They should do this as this is also in the journal submission guidelines:

<https://www.nature.com/ncomms/submit/how-to-submit>

"Isolated mass and percent yields should be reported at the end of each protocol."

Response to Reviewer 2

Reviewer's Comment:

I thank the authors for their second response, and will not debate subjective points further. In the first rebuttal the authors claimed to have collected elemental analysis and PXRD data for both Ce1 and Ce2 as requested. When I raised that I could not locate these data in the resubmission, the authors now respond that they have not collected any elemental analysis data for either complex, and only provide the PXRD of Ce1. In their first rebuttal the authors stated that complex Ce2 was a known compound already in the literature, and in their second response they now recognise that it is a new compound reported here for the first time. This inconsistency is confusing.

The authors state that they do not need elemental analysis or IR spectroscopy data, and provide a variety of reasons for the latter, including difficulties in collecting an IR spectrum due to a lack of a spectrometer in the nuclear facility and a dissatisfaction with the technique not providing meaningful information. The lack of ¹H NMR data for both complexes is attributed to poor solubility in the solvents investigated. No response is given for the lack of elemental analysis data for Ce2. The lack of PXRD data for Ce2 is attributed to sample degradation upon grinding, and in the authors' opinion is mitigated by the fact that only a structural comparison is needed. Responding to the above, I direct the authors to the journal guidelines:

<https://www.nature.com/ncomms/submit/chemical-characterisation>

“Yield and evidence of sample purity is required for each isolated compound.”

Whilst exceptions can (and should) be made for radiochemicals, these guidelines should apply to Ce1 and Ce2, and it is an expectation of the journal that the authors provide these data. I accept that a paucity of NMR data is common for paramagnetic complexes, and the solubility issues are noted. ¹H NMR spectroscopy should be attempted in a solvent that several mg of the complexes can dissolve in; can the authors please attempt a logical reasonable selection of solvents according to their judgement (e.g. d₂-DCM, d₅-chlorobenzene, d₈-THF, d₅-pyridine, d₃-MeCN). I note that the authors have stated that the complexes did not dissolve in any available NMR solvent, but it would be useful to know what was available and what has been attempted. In the absence of any solution NMR data a common basic requirement for reporting paramagnetic complexes in this referees experience is to provide elemental analyses and IR spectra. These are typically “cheaper” measurements to perform over other standard techniques, e.g. EPR spectroscopy and magnetometry, so they were suggested for perceived simplicity. To confirm the bulk purity of Ce2 the authors should provide elemental analysis results or other convincing evidence. The IR spectra were requested to mitigate the lack of NMR data provided; whilst accepting that these data are often not as useful, they can provide a fingerprint of molecular vibrational modes. If there are issues with collecting IR spectra in a nuclear facility is there any capacity to collect the IR spectra of Ce1 and Ce2 outside of this facility? These data should be provided if possible. This is a compromise from the initial request to collect SQUID and EPR data for Ce2, taking into account the authors' response.

The authors have also still not included the mass yields of Ce1 and Ce2 products. They should do this as this is also in the journal submission guidelines:

<https://www.nature.com/ncomms/submit/how-to-submit>

“Isolated mass and percent yields should be reported at the end of each protocol.”

Response: We apologize for the confusion and hope that our answers resolve any final concerns. We have collected ATR-IR spectra for Ce1 and Ce2 outside our facility and have added the spectra as well as a brief discussion in the ESI. The peaks are similar for the two compounds as expected with evidence of a para substituted phenyl ring breathing modes for Ce1 due to the substitution on the terpy ligand. We have also included CHN analysis from Midwest Microlabs in the synthesis section in the ESI. We have also included the deuterated solvents that the compounds were not soluble in. The mass of the target compounds associated with the yields obtained are also added to the synthesis section of the ESI.